# A Survey on Deep Learning for Theorem Proving

**Zhaoyu Li**[1], **Jialiang Sun**[1], **Logan Murphy**[1], **Qidong Su**[1], **Zenan Li**[2], **Xian Zhang**[3]
**Kaiyu Yang**[4*], **Xujie Si**[1,5]
[1]University of Toronto, [2]Nanjing University, [3]Microsoft Research Asia, [4]Meta FAIR,
[5]CIFAR AI Chair
{zhaoyu, six}@cs.toronto.edu

## Abstract

Theorem proving is a fundamental aspect of mathematics, spanning from informal reasoning in natural language to rigorous derivations in formal systems. In recent years, the advancement of deep learning, especially the emergence of large language models, has sparked a notable surge of research exploring these techniques to enhance the process of theorem proving. This paper presents a comprehensive survey of deep learning for theorem proving by offering (i) a thorough review of existing approaches across various tasks such as autoformalization, premise selection, proofstep generation, and proof search; (ii) an extensive summary of curated datasets and strategies for synthetic data generation; (iii) a detailed analysis of evaluation metrics and the performance of state-of-the-art methods; and (iv) a critical discussion on the persistent challenges and the promising avenues for future exploration. Our survey aims to serve as a foundational reference for deep learning approaches in theorem proving, inspiring and catalyzing further research endeavors in this rapidly growing field. A curated list of papers is available at `https://github.com/zhaoyu-li/DL4TP`.

## 1 Introduction

Proving theorems is a cornerstone of mathematics. Since the era of Euclid around 300 B.C., people have crafted theorems and proofs using a blend of natural language and mathematical symbols, meticulously evaluating their correctness through manual inspection. In the 1950s, a paradigm shift occurred with the exploration of computer-assisted proofs (Davis, 1957; Davis & Putnam, 1960), wherein a machine automatically applies deduction rules to prove assertions. These innovations laid the groundwork for the subsequent development of interactive theorem provers (Bruijn, de, 1970; Milner, 1972), enabling people to construct more intricate theorems and proofs by interacting with these systems. Building upon these advancements, later research extended the scope of theorem proving beyond mathematics, applying it to various practical applications such as software verification (Schumann, 2001) and hardware design (Kern & Greenstreet, 1999).

Exploring learning-based approaches for theorem proving has been a long-standing research focus, dating back to the 1990s (Suttner & Ertel, 1990; Denzinger et al., 1999). The recent development of deep learning, especially with the evolution of large language models (LLMs), has ignited a wave of research interest in this area again. As shown in Figure 1, the volume of papers on deep learning for theorem proving has grown approximately from 2 in 2016 to 45 in 2023, continuing to rise in 2024. However, despite such remarkable growth, this domain is characterized by a wide range of tasks, methods, datasets, and evaluations, which lack a cohesive framework to comprehend the true extent of progress and identify the underlying challenges and potential future work.

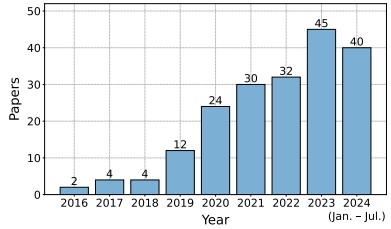

Figure 1: Papers on deep learning for theorem proving over the years (data for 2024 is up to July).

---

*Research conducted while Kaiyu Yang was at Caltech.

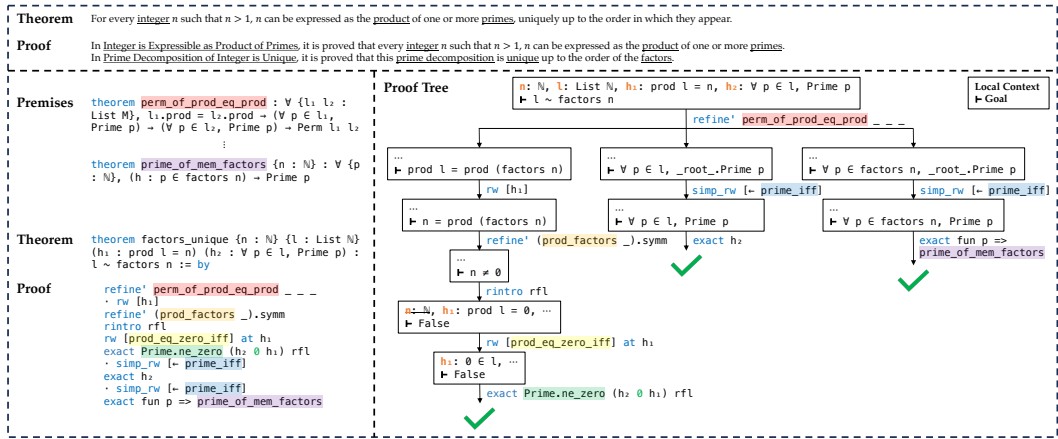

Figure 2: **Top:** The informal statement and proof of the Fundamental Theorem of Arithmetic in ProofWiki. **Bottom Left:** The formal statement and proof of the same theorem in the mathlib library (mathlib Community, 2020) of Lean 4. **Bottom Right:** The corresponding proof tree illustrating the formal proof process in Lean 4. Only changes in the local context of each node are marked for clarity. The references and premises used in the informal and formal proof are highlighted by underlines and colors respectively.

In this paper, we provide a comprehensive survey of more than 180 research papers in deep learning for theorem proving, aiming to map out the current research landscape and highlight key advancements systematically. We begin with the background for informal and formal settings of theorem proving (§2). Subsequently, we delve into the details of the tasks and methods within this domain (§3), which include autoformalization, premise selection, proofstep generation, proof search, and other tasks. We also review the datasets for theorem proving (§4), including manually curated and synthetically generated ones. Moreover, we examine the evaluation metrics and assess state-of-the-art performance (§5). Following this, we discuss the prevailing challenges and conclude with future directions (§6).

## 2 Background

In this section, we recall some fundamental concepts of theorem proving, including both informal and formal settings. Figure 2 shows an illustrative example of these two settings.

### 2.1 Informal Theorem Proving

Informal theorem proving involves establishing the truth of mathematical statements building on existing knowledge via intuitive reasoning and natural language explanations. This mirrors how people learn and prove theorems in everyday mathematics. For instance, to prove the Fundamental Theorem of Arithmetic in Figure 2 (Top), one needs to comprehend basic concepts like primes and might apply established results to draw a conclusion. Despite the ubiquity of informal theorem proving, as mathematics evolves, the theories and proofs tend to be more intricate, making verifying their correctness increasingly difficult.

### 2.2 Formal Theorem Proving

Formal theorem proving represents theorems and proofs in a machine-verifiable format, ensuring their correctness using rigorous logical rules. This field can be broadly classified into two paradigms: automated theorem proving (ATP) and interactive theorem proving (ITP).

ATP aims to verify formal statements fully automatically. Saturation-based theorem provers, including E (Schulz, 2002) and Vampire (Kovács & Voronkov, 2013), mainly operate on first-order logic (FOL) to autonomously generate logical consequences from a set of axioms until a proof or refutation is derived, or computational limits are reached. Similarly, geometric ATP systems such as GEX (Chou et al., 2000) prove geometry problems by iteratively applying

deduction rules. Other approaches, such as tableau-based methods like leanCoP (Otten & Bibel, 2003) and instantiation-based methods like iProver (Korovin, 2008), use other forms of proof calculi for proof construction. In addition to these, Boolean satisfiability (SAT) solvers (e.g., MiniSat (Eén & Sörensson, 2003), CaDiCaL (Biere, 2019)) and satisfiability modulo theories (SMT) solvers (e.g., Z3 (De Moura & Bjørner, 2008), CVC5 (Barbosa et al., 2022)) play a crucial role in ATP by efficiently handling propositional logic and theories such as arithmetic, bit-vectors, and arrays. Despite the sophisticated designs of these ATP systems, the inherent vast search space often limits their practicality in more complex problems.

In ITP, humans collaboratively prove theorems by interacting with *proof assistants*, such as Isabelle (Paulson, 1994), HOL Light (Harrison, 1996), Coq (Barras et al., 1999), Meta-math (Megill & Wheeler, 2019), and Lean (Moura & Ullrich, 2021). These proof assistants typically enable users to formalize theorems in higher-order logic and provide a language to build verifiable proofs. As shown in Figure 2 (Bottom Left), to prove a theorem (initial goal) in Lean, one can use *tactics* like `refine'` and `rw` as the proof steps. Applying a tactic either finishes the goal or decomposes it into simpler sub-goals, and the proof is complete when no further goals remain. When proving the current goal, one can apply assumptions in the local context and previously proven premises in the environment as tactic arguments. For example, the premise `perm_of_prod_eq_prod` is used as the argument of `refine'`. The proving process can be modeled as a proof tree, where each node is a proof state with a goal and its local context, and each edge is a tactic, as shown in Figure 2 (Bottom Right). Using proof assistants, researchers have successfully formalized and proved landmark theorems like the Four Color Theorem (Gonthier, 2008) and the Kepler Conjecture (Hales et al., 2017), and verified the correctness of critical software such as the seL4 microkernel (Klein et al., 2009) and the CompCert C compiler (Leroy et al., 2016). However, it is worth noting that these projects took several Ph.D. years to complete, requiring substantial labor and expertise.

## 3 Tasks and Methods

The emergence of deep learning has opened new avenues for the landscape of theorem proving, either enhancing or substituting traditional components involved in the process. This section categorizes and summarizes existing deep learning approaches into 5 tasks: autoformalization, premise selection, proofstep generation, proof search, and others.

### 3.1 Autoformalization

Autoformalization aims to convert informal theorems and proofs into machine-verifiable formats automatically. This task is notoriously challenging, requiring a profound understanding of semantics across informal and formal mathematics (Kaliszyk et al., 2014; 2015). Nonetheless, the success of autoformalization promises to facilitate the verification of mathematical papers and pave the way for general-purpose reasoning engines (Szegedy, 2020).

Wang et al. (2018; 2020) first explore using deep learning models in autoformalization. Inspired by the sequence-to-sequence models in neural machine translation (Sutskever et al., 2014; Cho et al., 2014), they experiment various encoder-decoder frameworks (Luong et al., 2017; Lample et al., 2018; Lample & Conneau, 2019) to convert LaTeX-written mathematical problem texts to the Mizar language (Rudnicki, 1992). Subsequent studies (Bansal & Szegedy, 2020; Cunningham et al., 2022) utilize similar neural architectures for HOL Light and Coq.

The recent development of LLMs and their in-context learning capabilities (Brown et al., 2020) have provided new opportunities for autoformalization. Some researchers study the potential of advanced LLMs using few-shot prompting techniques: Wu et al. (2022); Agrawal et al. (2022); Gadgil et al. (2022) leverage PaLM (Chowdhery et al., 2023) and Codex (Chen et al., 2021) to translate high school and undergraduate-level mathematical problems into Isabelle and Lean. LeanEuclid (Murphy et al., 2024) uses GPT-4 (Achiam et al., 2023) and GPT-4V to formalize both theorems and proofs in Euclidean geometry. Other research efforts (Jiang et al., 2023b; Patel et al., 2023; Zhao et al., 2024; Lu et al., 2024; Ying et al., 2024a; Poiroux et al., 2024) propose more structured approaches to autoformalization: For example, DSP (Jiang et al., 2023b) utilizes Minerva (Lewkowycz et al., 2022) to draft

informal proofs and map them into formal sketches, with ATP systems employed to fill in the missing details in the proof sketch. Zhao et al. (2024) improves informal proofs and formal sketches in DSP with sub-goal proofs and prompt selection respectively. Additionally, a line of research (Azerbayev et al., 2023; Jiang et al., 2023a; Azerbayev et al., 2024; Shao et al., 2024; Ying et al., 2024b) focuses on training LLMs on large-scale datasets containing both informal and formal mathematical data to evaluate their performance on autoformalization. Recent studies (Liu et al., 2023; Pan et al., 2023; Olausson et al., 2023; Ye et al., 2023; Zhou et al., 2024a; Huang et al., 2024; Xin et al., 2024b; Jiang et al., 2024; Quan et al., 2024; Xin et al., 2024a) also apply autoformalization as a key step in various downstream tasks. For instance, SAT-LM (Ye et al., 2023) uses LLMs to formalize natural language problems using declarative prompting and solve them using Z3 on several reasoning tasks. DTV (Zhou et al., 2024a) leverages autoformalization to ground LLM reasoning, formalizing LLM-generated answers and verifying them with ATP tools. Besides these efforts, Wu et al. (2022); Azerbayev et al. (2023); Jiang et al. (2023a); Lu et al. (2024) explore advanced LLMs like Codex and GPT-4 for informalization, i.e., the translation of formal statements into natural language.

## 3.2 Premise Selection

Given a large collection of previously proven lemmas, premise selection is to retrieve the helpful lemmas that can contribute to a successful proof. It is an enduring challenge in both mathematical research and ATP/ITP systems (Kühlwein et al., 2012; Alama et al., 2014).

The seminal works (Irving et al., 2016; Kaliszyk et al., 2017) model premise selection as a binary classification task, embedding theorems and premises with a variety of neural networks including convolutional neural networks (CNNs), recurrent neural networks (RNNs), and hybrid models. These embeddings are then combined to feed into a logit layer to predict their relevance. Follow-up works (Kucik & Korovin, 2018; Bansal et al., 2019; Piotrowski & Urban, 2020a; Proroković et al., 2021; Szegedy et al., 2021) extend previous frameworks by using a better representation of features or more sophisticated architectures like Wavenet (Van Den Oord et al., 2016) and Transformer (Vaswani et al., 2017).

Given the inherent structured nature of mathematical formulas, a stream of research (Wang et al., 2017; Peng & Ma, 2017; Olšák et al., 2019; Goertzel & Urban, 2019; Crouse et al., 2019; Paliwal et al., 2020; Rawson & Reger, 2020; Liu et al., 2022a; Goertzel et al., 2022; Holden & Korovin, 2023; Jakubüv et al., 2023) parses formal statements into trees or graphs and leverages tree-structured neural networks (Tai et al., 2015) or graph neural networks (GNNs) (Duvenaud et al., 2015; Veličković et al., 2018; Xu et al., 2019) for encoding. For example, FormulaNet (Wang et al., 2017) proposes a graph embedding method that preserves the information of edge ordering. Olšák et al. (2019) introduces a GNN framework that captures several logical invariances in FOL formulas. Paliwal et al. (2020) conducts comprehensive experiments to evaluate various designs for the graph representations of formulas in HOL Light. Subsequent works (Li et al., 2021b; Lin et al., 2021) explore graph contrastive learning (Oord et al., 2018; Chen et al., 2020) to train GNNs for premise selection. Moreover, Ferreira & Freitas (2020b); Bauer et al. (2023) construct a dependency graph over a large corpus by representing theorems and premises as nodes and their dependencies as edges, and leverage GNNs to predict the link between nodes for premise selection.

With the advancement of pre-trained language models, some efforts (Ferreira & Freitas, 2020a; Welleck et al., 2021) fine-tune BERT (Devlin et al., 2019)-like models to embed natural language statements into vectors and select premises using a linear classifier layer. Later works (Ferreira & Freitas, 2021; Tran et al., 2022; Trust et al., 2022; Dastgheib & Asgari, 2022; Yeh et al., 2023; Yang et al., 2023) leverage different pre-trained models (Liu et al., 2019; Song et al., 2020; Xue et al., 2022) for encoding and retrieve informal/formal premises based on several similarity metrics. Specifically, ReProver (Yang et al., 2023) trains its retriever based on dense passage retrieval (DPR) (Karpukhin et al., 2020) to select premises in Lean. Han et al. (2021) also explores fine-tuning over large informal mathematical corpora using the contrastive objective (Oord et al., 2018), while PACT (Han et al., 2022) uses the auto-regressive objective for formal premise selection. Additionally, several research (Kovriguina et al., 2022; Tworkowski et al., 2022; Mikuła et al., 2024) design a second phase to re-rank the selected subset of premises, enabling a more accurate selection.

### 3.3 Proofstep Generation

Proofstep generation is the core problem for theorem proving, which aims to predict one or more steps to build the proof of a theorem. This task also refers to tactic prediction in ITP, which has been widely studied in tactic-based ATP systems (hammers) (Böhme & Nipkow, 2010; Blanchette et al., 2016; Czajka & Kaliszyk, 2018).

A stream of research (Whalen, 2016; Huang et al., 2019; Bansal et al., 2019; Paliwal et al., 2020; Sanchez-Stern et al., 2020; Wu et al., 2021b; Rute et al., 2024) treats tactic prediction as a classification problem and uses separate neural networks to predict the tactic and its arguments. For example, Gamepad (Huang et al., 2019) employs TreeLSTM (Tai et al., 2015) to encode the proof states and two distinct linear layers for tactic and argument prediction. Proverbot9001 (Sanchez-Stern et al., 2020) uses a feed-forward neural network and an RNN to predict the tactic and its arguments respectively. Besides these works, ASTactic (Yang & Deng, 2019) proposes a decoder that generates the tactic as a program, using an RNN to control the generation based on a predefined context-free grammar. Later studies (First et al., 2020; First & Brun, 2022; Sanchez-Stern et al., 2023) improve ASTactic by incorporating prior proof scripts, combining varied models, and modeling identifiers of theorems.

Subsequent advancements (Polu & Sutskever, 2020; Polu et al., 2023; Han et al., 2022; Jiang et al., 2021; Zhang et al., 2023a; Yeh et al., 2023; Xiong et al., 2023; Wang et al., 2023a; Vishwakarma & Mishra, 2023; Gloeckle et al., 2023; First et al., 2023; Xin et al., 2024a; Wang et al., 2024; Lin et al., 2024a; Wu et al., 2024) formulate tactic prediction as language modeling. Specifically, GPT-$f$ (Polu & Sutskever, 2020) first apply a conditional language modeling objective to train decoder-only Transformers to generate a proof step in the format of `GOAL <GOAL> PROOFSTEP <PROOFSTEP>`. Baldur (First et al., 2023) applies a similar objective to generate or repair the whole proof at once. POETRY (Wang et al., 2024) introduces a level-by-level approach, recursively generating a formal sketch of the proof at each level and solving the current level's theorem or conjecture until the proof is complete. Several studies (Szegedy et al., 2021; Tworkowski et al., 2022; Welleck et al., 2022; Jiang et al., 2022; Yang et al., 2023) also jointly train tactic prediction with premise selection. For instance, NaturalProver (Welleck et al., 2022) trains GPT-3 (Brown et al., 2020) with constrained decoding to encourage using retrieved references in the proof steps. Thor (Jiang et al., 2022) adds a `<hammer>` token to learn when to invoke a hammer system (Böhme & Nipkow, 2010) for premise selection to simplify the proof. In the geometry domain, Chen et al. (2022); Liang et al. (2023); Trinh et al. (2024) follow the same paradigm, auto-regressively generating the proof sequence at each step. Notably, AlphaGeometry (Trinh et al., 2024) trains a decoder-only Transformer to predict the auxiliary constructions in the proofs of International Mathematical Olympiad (IMO) geometry problems. Besides training on proof data, Azerbayev et al. (2024); Shao et al. (2024); Ying et al. (2024b) train LLMs on extensive general mathematical corpora and evaluate their abilities for generating formal proofs.

With the development of LLMs, some researchers (Zhang et al., 2023b; Yousefzadeh & Cao, 2023; Scheidt, 2023; Frieder et al., 2023a;b;c; Zhang et al., 2024; Poulsen et al., 2024) also explore prompting state-of-the-art LLMs without any additional training to generate proofs across various domains. For example, Frieder et al. (2023b) evaluates the performance of ChatGPT and GPT-4 on completing informal mathematical proofs, and Selene (Zhang et al., 2024) focuses on project-level automated proof in software verification based on the seL4 project. Recent explorations further propose more structured pipelines for formal proof generation (Jiang et al., 2023b; Zhao et al., 2024; Zheng et al., 2024; Xin et al., 2024b; Huang et al., 2024; Thakur et al., 2024): For instance, Lyra (Zheng et al., 2024) leverages two correction strategies, namely tool correction and conjecture correction, as post-processing and error feedback mechanisms to refine incorrect proofs generated by GPT-4.

### 3.4 Proof Search

Proof search seeks to systematically traverse the vast landscape of potential proof paths to construct a valid proof tree for a given theorem in formal systems. It is not only a long-standing research focus in ATP (Urban et al., 2011; Kaliszyk & Urban, 2015a; Jakubův & Urban, 2017) but also a vital process for tactic-based models to complete the proof in ITP.

A thread of research treats branching as a classification task and trains deep learning models on successful proof paths in a supervised fashion to guide the search in various ATP systems. Specifically, a large body of works (Loos et al., 2017; Chvalovskỳ et al., 2019; Jakubův & Urban, 2019; Aygün et al., 2020; Jakubův et al., 2020; Suda, 2021b; Chvalovskỳ et al., 2021; Goertzel et al., 2021; Suda, 2021a; Firoiu et al., 2021; Aygün et al., 2022; Goertzel et al., 2022; Jakubův et al., 2023; Bártek & Suda, 2023) exploit various RNNs, GNNs, or hybrid models for the clause selection in saturation-based provers. Similarly, Piepenbrock et al. (2022b); Chvalovskỳ et al. (2023) and Piotrowski & Urban (2020b) focus on guiding instantiation and connection tableau respectively. Some works (Rawson & Reger, 2019; Olšák et al., 2019; Rawson & Reger, 2021; Zombori et al., 2021b; Wei et al., 2024) further combine the supervised trained models as the policy or value networks to guide the Monte Carlo Tree Search (MCTS) or A* search across various ATP systems. Additionally, another direction of research (Kusumoto et al., 2018; Fawzi et al., 2019; Abdelaziz et al., 2020; Zombori et al., 2021a; Crouse et al., 2021; Piepenbrock et al., 2021; 2022a; Liu et al., 2022b; Abdelaziz et al., 2022; Fokoue et al., 2023; McKeown & Sutcliffe, 2023; Shminke, 2023) models proof search as a Markov decision process and applies reinforcement learning (RL) to train and guide the proof search. For instance, TRAIL (Crouse et al., 2021) uses the policy gradient (Sutton et al., 1999) to train an attention-based action policy in saturation-based provers, and Fawzi et al. (2019) applies deep Q-learning (Mnih et al., 2013) to guide the choice of inference rules in a semi-algebraic proof system (Lovász & Schrijver, 1991) for polynomial inequalities.

Most tactic-based models in ITPs use beam search to sample multiple tactic predictions per step with breadth-first (Bansal et al., 2019), depth-first (Yang & Deng, 2019), or best-first heuristics (Polu & Sutskever, 2020) to traverse the search space. In particular, GPT-$f$ (Polu & Sutskever, 2020) and FMSCL (Polu et al., 2023) train language models with outcome and proof size objectives as value functions to perform the best-first search. Besides these methods, Whalen (2016); Mo et al. (2020); Gauthier (2020); Wu et al. (2021a); Gauthier (2021); Lample et al. (2022); Wang et al. (2023a); Brandfonbrener et al. (2024) combine MCTS or use RL to train and guide the search procedure. For example, TacticZero (Wu et al., 2021a) employs the policy gradient to jointly learn tactic prediction and proof search, HTPS (Lample et al., 2022) adopts an AlphaZero (Silver et al., 2018)-like approach with online training, and DT-Solver (Wang et al., 2023a) improves MCTS with dynamic tree sampling and proof-level value function. Additionally, COPRA (Thakur et al., 2024) implements a language-agent method that uses GPT-4 to perform a backtracking search based on the proof history, and TrialMaster (An et al., 2024) finetunes LLMs with trial-and-error data to do backtracking.

## 3.5 Other Tasks

In addition to the primary tasks outlined previously, we briefly list several other prediction tasks that are related or helpful to theorem proving. One prominent line of research (Clark et al., 2020; Saha et al., 2020; Dalvi et al., 2021; Tafjord et al., 2021; Sanyal et al., 2022; Bostrom et al., 2022; Hong et al., 2022; Mishra et al., 2022; Yang et al., 2022; Tafjord et al., 2022; Morishita et al., 2023; Saparov & He, 2023) focuses on generating step-by-step rationale of a hypothesis from a set of known facts, which answers an open-domain question. These works primarily perform FOL rule deduction over natural language, in the form of an entailment tree, an analogy to the proof tree in formal theorem proving. Another area of interest (Urban & Jakubův, 2020; Piotrowski & Urban, 2020b; Rabe et al., 2021; Johansson & Smallbone, 2023; Bengio & Malkin, 2024; Poesia et al., 2024) is automated conjecturing, which aims to discover new and interesting theorems beyond existing data. Additionally, research efforts such as Huang et al. (2019); Glorot et al. (2019); Polu et al. (2023) predict the proof length remained for a goal. Lee et al. (2020); Wu & Wu (2021) investigate proving theorems in the latent space. IsarStep (Li et al., 2021a) predicts the intermediate proposition given surrounding proofs. Skip-tree (Rabe et al., 2021) and PACT (Han et al., 2022) propose several self-supervised tasks by masking various proof terms for training language models. LIME (Wu et al., 2021c) creates three synthetic pre-training tasks inspired by three reasoning primitives of deduction, induction, and abduction, to improve the performance of formal theorem proving. Recently, Li et al. (2023) proposes to match the informal proofs with theorem statements from a large database, REFACTOR (Zhou et al., 2024b) and ATG (Lin et al., 2024c) focus on extracting or generating new useful formal theorems from proofs.

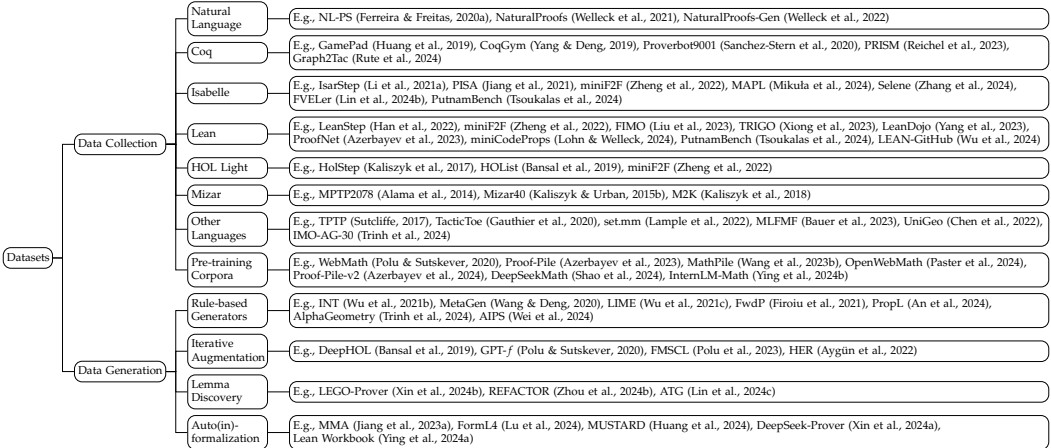

Figure 3: The taxonomy of datasets in theorem proving.

## 4 Datasets

This section classifies and summarizes datasets for theorem proving into 2 categories: i) datasets extracted from existing corpora or manually curated and ii) those using synthetic generation or augmentation methods. The taxonomy of these datasets is shown in Figure 3, and Table 1 in the Appendix provides more detailed information.

### 4.1 Data Collection

We begin with the review of informal datasets. NL-PS (Ferreira & Freitas, 2020a) first builds a natural language premise selection dataset source from ProofWiki. Similarly, NaturalProofs (Welleck et al., 2021) further incorporates theorems from Stacks, and textbooks for premise selection. Adapted from NaturalProofs, NaturalProofs-Gen (Welleck et al., 2022) utilizes a subset of theorems and proofs for informal proof generation.

For formal datasets, a line of efforts has been made to extract and clean theorems and proofs from established formal libraries and verification projects. Notable datasets for Coq include GamePad (Huang et al., 2019), CoqGym (Yang & Deng, 2019), Proverbot9001 (Sanchez-Stern et al., 2020), PRISM (Reichel et al., 2023), and Graph2Tac (Rute et al., 2024), which are constructed based on mathematical or software verification projects. For Isabelle, datasets like IsarStep (Li et al., 2021a), PISA (Jiang et al., 2021), and MAPL (Mikuła et al., 2024) are built on the Archive of Formal Proofs and Isabelle Standard Library, while Selene (Zhang et al., 2024) and FVELer (Lin et al., 2024b) are constructed based on the seL4 project (Klein et al., 2009). LeanStep (Han et al., 2022), LeanDojo (Yang et al., 2023), MLFMF (Bauer et al., 2023), and LEAN-GitHub (Wu et al., 2024) utilize Lean's open-source libraries (e.g., the mathlib library (mathlib Community, 2020)). Datasets for other proof assistants include Hol-Step (Kaliszyk et al., 2017) and HOList (Bansal et al., 2019) for HOL Light, MPTP2078 (Alama et al., 2014), Mizar40 (Kaliszyk & Urban, 2015b), and M2K (Kaliszyk et al., 2018) for Mizar, etc. Besides extracting data from existing projects, several works manually formalize or annotate the problems into formal languages: miniF2F (Zheng et al., 2022), FIMO (Liu et al., 2023), ProofNet (Azerbayev et al., 2023), and PutnamBench (Tsoukalas et al., 2024) manually formalize high school and college-level problems in mathematical competitions or textbooks in Lean or Isabelle, and miniCodeProps (Lohn & Welleck, 2024) translates Haskell programs into Lean. For other domains, TRIGO (Xiong et al., 2023) formalizes the trigonometric reduction problem in Lean, while UniGeo (Chen et al., 2022) and IMO-AG-30 (Trinh et al., 2024) annotate proof steps for geometry proving problems in their designed formal languages.

On the other hand, a large body of recent studies leverages large-scale online corpora with billions of tokens from informal and formal mathematical data to build the pre-training datasets that could aid in theorem proving. These datasets include WebMath (Polu & Sutskever, 2020), Proof-Pile (Azerbayev et al., 2023), DeepSeekMath (Shao et al., 2024), etc.

### 4.2 Data Generation

Beyond utilizing existing projects, researchers also study the generation of new theorems and proofs. A line of work (Wu et al., 2021b;c; Firoiu et al., 2021; Trinh et al., 2024; An et al., 2024; Wei et al., 2024) develops rule-based generators to produce new data by iteratively sampling inference rules from a pre-defined set. This process enables manual control over both the quantity and difficulty of the generated theorems. For example, INT (Wu et al., 2021b) produces 1.5 million inequality training problems with different proof lengths and axiom distributions than the testing ones, while AlphaGeometry (Trinh et al., 2024) synthesizes over 100 million training data with proof lengths ranging from 1 to 247. MetaGen (Wang & Deng, 2020) also trains a neural generator to produce theorems similar to human-write ones.

Alternative approaches turn to iteratively augment the training dataset with fixed theorems but newly generated proofs. A line of work (Bansal et al., 2019; Polu & Sutskever, 2020; Polu et al., 2023) adopts the idea of expert iteration (Silver et al., 2018), which repeatedly applies the trained prover on existing theorems and adds the successful proof paths as new data points to train the prover further. Aygün et al. (2022) also proposes to adapt hindsight experience replay (Andrychowicz et al., 2017) to FOL provers, which leverages previously unsuccessful proof trajectories by viewing their final states as the desired ones. Meanwhile, a line of RL-based methods can also be viewed in this category.

Additionally, some studies aim to generate intermediate helpful lemmas. REFACTOR (Zhou et al., 2024b) trains a GNN to extract lemmas from proofs, which can be used to streamline the proofs of other theorems. Similarly, ATG (Lin et al., 2024c) generates new theorems based on MCTS and self-play learning to shorten proofs. LEGO-Prover (Xin et al., 2024b) prompts GPT-4 to generate sub-goal lemmas for the proof of a theorem and finalize it by proving or retrieving these lemmas. Newly proven lemmas are added to a library for future use. Works on conjecturing are also relevant to this field, although the generated conjectures could be more challenging than existing theorems and may be incorrect or difficult to prove.

Moreover, several recent approaches leverage auto(in)formalization for data generation. Using GPT-4, MMA (Jiang et al., 2023a) informalizes all theorem statements in Archive of Formal Proofs and mathlib, while FormL4 (Lu et al., 2024) generates informal descriptions of both theorems and proofs extracted from mathlib. MUSTARD (Huang et al., 2024) synthesizes problems from a few sampled seed concepts, crafts informal proofs, and translates them into Lean to verify their correctness. DeepSeek-Prover (Xin et al., 2024a) autoformalizes mathematical competition problems and filters high-quality statements through model scoring and hypothesis rejection methods. Similarly, Lean Workbook (Ying et al., 2024a) proposes an active learning pipeline that autoformalizes mathematical questions and filters them through Lean's compilation, LLMs' evaluation, and human diagnostics at each round.

## 5 Evaluations

In this section, we focus on the evaluations of deep learning approaches in theorem proving, analyzing the key metrics and state-of-the-art performance for each task. Detailed accuracies of state-of-the-art methods on several key datasets are provided in Table 2 in the Appendix.

**Autoformalization.** The assessment of autoformalization mainly relies on manually checking the equivalence between informal and formalized statements. Recent studies (Wu et al., 2022; Azerbayev et al., 2023) indicate that state-of-the-art LLMs with few-shot prompting can only correctly formalize 25% and 13% high-school and undergraduate-level problems, highlighting the challenge in autoformalization. In addition, both studies reveal that LLMs exhibit considerably higher efficacy in informalization, achieving accuracies of around 76% and 62%, respectively. Despite the modest success in autoformalizing statements, subsequent research (Jiang et al., 2023b; Zhao et al., 2024; Xin et al., 2024b) reveals the benefit of autoformalizing intermediate sub-goals to generate formal proofs in modularized pipelines.

**Premise Selection.** Retrieval metrics are widely used as the evaluation metric for premise selection. For example, recall at $k$ (R@$k$) measures the ratio of correctly used premises in ground-truth proof within the top-$k$ selections, and the mean reciprocal rank (MRR)

computes the average reciprocal rank of the first correctly selected premise. Among existing methods, DPR with a Transformer encoder has significantly improved upon traditional methods, demonstrating a remarkable generalization ability to unseen data for premise selection. For example, ReProver (Yang et al., 2023) achieves 27.6% for R@10 and 0.24 for MRR on the LeanDojo benchmark for retrieving unseen premises in training, while the baseline method BM25 (Robertson et al., 2009) achieves 15.5% for R@10 and 0.14 for MRR. Furthermore, a DPR-based retriever, Magnushammer (Mikuła et al., 2024), outperforms the classic hammer system, Sledgehammer (Böhme & Nipkow, 2010), when used to select premises for the Thor prover (Jiang et al., 2022), improving the theorem proving success rate from 57% to 71% on the PISA dataset and from 28.3% to 36.9% on the miniF2F-valid dataset.

**Theorem Proving.** The effectiveness of proofstep generation, proof search, and support from autoformalization and premise selection can be collectively evaluated by their success rate in proving theorems within a test set. Recent research (Zheng et al., 2024; Xin et al., 2024b) shows impressive performance increases using state-of-the-art LLMs like GPT-4 in structured frameworks than fine-tuned tactic-based language models with search heuristics. For instance, LEGO-Prover (Xin et al., 2024b) achieves 57.0% accuracy and 50.0% accuracy on the valid and test set of miniF2F, while the previous best prover Thor (Jiang et al., 2022) with expert iteration training (Wu et al., 2022) achieves 37.3% and 35.2%. Moreover, advanced learning-based proof searches could further improve the performance of ITP/ATP systems. HTPS (Lample et al., 2022) achieves an accumulative successful rate of 58.6% on miniF2F-valid through online training and a 41.0% success rate with 64 search attempts on miniF2F-test. Besides, the RL-based ATP system NIAGRA (Fokoue et al., 2023) outperforms both E (Schulz, 2002) and Vampire (Kovács & Voronkov, 2013) on the MPTP2078 dataset.

**Caveats.** Tasks related to theorem proving can be tricky to evaluate. For autoformalization, manual evaluation is costly, whereas most automated metrics are inaccurate. For example, the compilation rate evaluates only syntactic correctness, and the BLEU score (Papineni et al., 2002) struggles with formalizations semantically similar but not logically equivalent to the ground truth. Although LeanEuclid (Murphy et al., 2024) provides automatic semantic evaluation between the autoformalized theorem statement and the ground truth, its domain is limited to Euclidean geometry, making it difficult to generalize to other areas. For premise selection, metrics rely on the premises that are used in ground-truth proofs, so they may neglect other valid premises for alternative correct proofs different from the ground-truth ones, causing false negatives. Additionally, the evaluation of theorem proving is further complicated by the variety of experimental setups, as detailed in §6.1.

# 6 Discussions

## 6.1 Challenges

Despite significant progress, deep learning for theorem proving still faces many challenges, including data scarcity, disunified evaluation protocols, and human-AI interaction.

**Data Scarcity.** The amount of formal proof data is growing but is still far behind other domains where LLMs are successful, e.g., code generation. The largest corpora of Isabelle proofs, Archive of Formal Proofs, currently contains 250K proofs. Lean's mathlib contains 140K proofs. This amount of data is decent for small models (e.g., billions of parameters) but insufficient for models with hundreds of billions of parameters. Although the use of rule-based generators could offer some assistance, the complexity and quality of the generated data often do not match that of human-written ones. Furthermore, autoformalization is even more data-scarce, due to the difficulty in obtaining aligned informal-formal pairs.

**Evaluation.** Compared to traditional deep learning tasks such as classification, it is considerably more complex to evaluate the performance of theorem provers comprehensively. Firstly, results across different proof assistants are not directly comparable. Even though miniF2F is available across multiple proof assistants, the impact of proof automation tools (e.g., Isabelle has Sledgehammer (Böhme & Nipkow, 2010) whereas Lean does not) often outweighs the differences attributable to deep learning models. Secondly, resource constraints during evaluation, such as time and the number of attempts, can significantly affect performance

and the relative ranking of different methods. While more attempts may favor larger models like LLMs, tight time constraints, especially in real-time applications, may favor simpler, faster models. Without a specific application as the context, it is unclear what evaluation setting makes the most sense. Thirdly, the use of pre-trained LLMs like GPT-4 may introduce data contamination issues (Rute et al., 2024), as these models might have been pre-trained on existing informal or formal testing problems, leading to unfair comparisons. We as a community still lack a systematic evaluation framework across different types of neural theorem provers, despite nascent efforts (Lamont et al., 2024; Rute et al., 2024).

**Human-AI Interaction.** One motivation for theorem proving is to assist human mathematicians (Castelvecchi, 2021; Sørensen et al., 2021). However, existing research has led to surprisingly few tools useful for them. Current methods are evaluated following standard deep learning protocols: running neural networks or querying LLMs in a Python program, testing the prover on a dataset, and calculating the percentage of successfully proved ones. This practice is misaligned with the needs of mathematicians. First, mathematicians need a tool that can be called easily in a proof assistant. Second, the tool must run on consumer CPUs with low latency. Third, instead of a performance measure, mathematicians care more about whether the prover can help with the specific theorems they are working on. These theorems are often out of the training distribution and pose a challenge for the prover to generalize to other domains. Although initial efforts have been made to develop user-oriented tools (Welleck & Saha, 2023; Song et al., 2024; Rute et al., 2024), there is abundant room to improve the user experience and explore other forms of interaction, which requires close collaboration between deep learning researchers and mathematicians (Collins et al., 2023).

## 6.2 Future Directions

Combining deep learning, especially LLMs, with theorem proving provides a promising avenue for enhancing the mathematical capabilities of AI and may significantly impact various disciplines. We conclude our survey paper by listing a few future directions we are particularly excited about, envisioning significant strides in these burgeoning domains:

**Conjecturing.** Beyond merely proving theorems, mathematicians would always explore theories in a domain, identify underlying problem structures, and formulate new conjectures. These explorative activities around conjecturing are indispensable for mathematicians but are relatively limited in current deep learning approaches (Urban & Jakubüv, 2020; Johansson & Smallbone, 2023; Bengio & Malkin, 2024). By enabling AI to generate useful conjectures, it can explore the space of mathematics autonomously. When combined with theorem proving, such exploration can also be used to discover new mathematical knowledge. Furthermore, a direct application of conjecturing is to generate more theorem (and proof) data, which could mitigate the data scarcity inherent in theorem proving.

**Verified Code Generation.** As AI coding assistants such as GitHub Copilot become prevalent, it is increasingly important to be able to verify LLM-generated code. Proof assistants, especially Coq, have been widely used for software verification (Leroy et al., 2016; Gu et al., 2016). Therefore, methods for theorem proving surveyed in this paper could potentially play a role in generating verified code. There is a plethora of problems to explore in this space. For example, one can train or prompt LLMs to synthesize program invariants (Pei et al., 2023; Kamath et al., 2023) or generate programs in verification-friendly languages such as Dafny (Sun et al., 2023), Verus (Yao et al., 2023), and F* (Chakraborty et al., 2024).

**Math Education.** A roadblock to democratizing math education is the lack of qualified tutors to provide feedback to students (Kumar et al., 2023). Formal mathematics can potentially mitigate this issue by providing an environment for students to explore and receive automatic and reliable feedback. AI has demonstrated promise in guiding students in this process. For example, Buzzard (2024) reported that Lean Copilot (Song et al., 2024) effectively assisted in proving a bunch of problems in his undergraduate course and solved questions on the Lean Zulip. To further integrate AI-driven formal tutoring into mainstream education, informalization is essential to make formal proofs accessible to students unfamiliar with formal languages. Looking ahead, we believe that theorem proving with LLMs will pave the way for intelligent tutors, enhancing math education for a broader audience.

**Acknowledgments**

We thank the anonymous reviewers for their insightful comments. This work was supported, in part, by Individual Discovery Grants from the Natural Sciences and Engineering Research Council of Canada, and the Canada CIFAR AI Chair Program.

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

# A    Lists of Datasets and State-of-the-Art Approaches

| Dataset | Language[1] | Size[2] | Source | Task[3] |
|---|---|---|---|---|
| NL-PS (Ferreira & Freitas, 2020a) | NL | 20,401 | ProofWiki | PS$_1$ |
| NaturalProofs (Welleck et al., 2021) | NL | 48,783 | ProofWiki, Stacks, textbooks | PS$_1$ |
| NaturalProofs-Gen (Welleck et al., 2022) | NL | ~33K | NaturalProofs | PG |
| GamePad (Huang et al., 2019) | Coq | 1,602 | Feit-Thompson theorem | PG |
| CoqGym (Yang & Deng, 2019) | Coq | 70,856 | 123 open-source projects from Github | TP |
| Proverbot9001 (Sanchez-Stern et al., 2020) | Coq | 501* | CompCert | TP |
| Graph2Tac (Rute et al., 2024) | Coq | ~520K | 120 open-source packages from opam | TP |
| PISA (Jiang et al., 2021) | Isabelle | ~183K | Achieve of Formal Proofs, Isabelle standard library | TP |
| MAPL (Mikuła et al., 2024) | Isabelle | ~433K | Achieve of Formal Proofs, Isabelle standard library | PS$_1$ |
| Selene (Zhang et al., 2024) | Isabelle | 360* | seL4 | TP |
| FVELer (Lin et al., 2024b) | Isabelle | 29,125 | seL4 | TP |
| LeanStep (Han et al., 2022) | Lean | ~20K | Lean core library, mathlib | TP |
| TRIGO (Xiong et al., 2023) | Lean | 427 | trigonometry problems from tiku | TP |
| LeanDojo (Yang et al., 2023) | Lean | 98,734 | mathlib | TP |
| MLFMF (Bauer et al., 2023) | Lean, Agda | 270,647 | mathlib, Agda standard library, UniMath, TypeTopology | PS$_1$ |
| miniCodeProps (Lohn & Welleck, 2024) | Lean, Haskell | 177* | Tons of Inductive Programs | TP |
| LEAN-GitHub (Wu et al., 2024) | Lean | 28,597 | 147 repositories on the web | TP |
| FIMO (Liu et al., 2023) | NL, Lean | 149* | IMO Shortlisted Problems | AF |
| ProofNet (Azerbayev et al., 2023) | NL, Lean | 371* | undergraduate textbooks | AF |
| miniF2F (Zheng et al., 2022) | NL, Lean, Isabelle, Metamath, HOL Light | 488* | IMO, AIME, AMC, MATH, custom | TP |
| PutnamBench (Tsoukalas et al., 2024) | NL, Lean, Isabelle | 640* | William Lowell Putnam Mathematical Competition | TP |
| HolStep (Kaliszyk et al., 2017) | HOL Light | 11,410 | multivariate analysis library & Kepler conjecture | PS$_1$ |
| HOList (Bansal et al., 2019) | HOL Light | 31,662 | Kepler conjecture | TP |
| MPTP2078 (Alama et al., 2014) | Mizar, TPTP | 2,078 | Mizar Mathematical Library | PS$_1$, PS$_2$ |
| Mizar40 (Kaliszyk & Urban, 2015b) | Mizar, TPTP | ~58k | Mizar Mathematical Library | PS$_1$, PS$_2$ |
| M2K (Kaliszyk et al., 2018) | Mizar, TPTP | 2,003 | Mizar40 | PS$_2$ |
| TPTP (Sutcliffe, 2017) | TPTP | 11,395 | ATP system evaluation | PS$_2$ |
| TacticToe (Gauthier et al., 2020) | HOL4 | 7,164* | HOL4 standard library | TP |
| set.mm (Lample et al., 2022) | Metamath | 37,091 | set.mm | TP |
| UniGeo (Chen et al., 2022) | NL, DSL | 9,543 | high school geometry problems from IXL | PG |
| IMO-AG-30 (Trinh et al., 2024) | DSL | 30* | IMO geometry problems | TP |
| INT (Wu et al., 2021b) | DSL | 1,501K | synthetic inequality problems | TP |
| AlphaGeometry (Trinh et al., 2024) | DSL | ~100M | synthetic geometry problems | TP |
| MMA (Jiang et al., 2023a) | NL, Lean, Isabelle | 332,774 | autoinformalization from Achieve of Formal Proofs and mathlib | AF |
| FormL4 (Lu et al., 2024) | NL, Lean | 17,461 | autoinformalization from mathlib, Arithmo test set | AF |
| MUSTARD (Huang et al., 2024) | NL, Lean | 5,866 | autoformalization from synthetic problems | TP |
| DeepSeek-Prover (Xin et al., 2024a) | NL, Lean | 8,066,621 | autoformalization from competition problems | TP |
| Lean Workbook (Ying et al., 2024a) | NL, Lean | ~57K | autoformalization from Art of Problem Solving | AF |
| WebMath (Polu & Sutskever, 2020) | mixed | 35B | GitHub, arXiv, Stack Exchange | PT |
| Proof-Pile (Azerbayev et al., 2023) | mixed | 8.3B | arXiv, Stack Exchange, formal libraries, ProofWiki, Wikipedia, books, MATH | PT |
| MathPile (Wang et al., 2023b) | mixed | 9.5B | arXiv, textbooks, Stack Exchange, Wikipedia, ProofWiki, Web | PT |
| OpenWebMath (Paster et al., 2024) | mixed | 14.7B | forum posts, educational content, reference pages, scientific papers, blogs, and others | PT |
| Proof-Pile-v2 (Azerbayev et al., 2024) | mixed | 55B | Github, Stack, formal proof steps, OpenWebMath, arXiv | PT |
| DeepSeekMath (Shao et al., 2024) | mixed | 120B | OpenWebMath, Web | PT |
| InternLM-Math (Ying et al., 2024b) | mixed | 125B | Knowledge Pile, Proof-Pile-v2, synthetic data | PT |

Table 1: Summary of existing datasets for theorem proving. For datasets derived from generation methods, we primarily highlight several key examples here.

| Language | Dataset | Best Accuracy (%) | Evaluation Metric | Method |
|---|---|---|---|---|
| Coq | CoqGym | 33.8 | timeout of 600s | Diva (First & Brun, 2022) + CoqHammer (Czajka & Kaliszyk, 2018) |
| Lean | LeanDojo (random) | 57.7 | pass@1 + timeout of 600s | temperature scaling (Gloeckle et al., 2023) |
| | ProofNet (valid/test) | 49.2/53.2 | top-1 accuracy@50 | type checking (Poiroux et al., 2024) |
| | miniF2F (valid/test) | 60.2/46.3 | cumulative/pass@64 | DeepSeek-Prover (Xin et al., 2024a) |
| Isabelle | miniF2F (valid/test) | 57.0/51.2 | pass@100/pass@200 | LEGO-Prover (Xin et al., 2024b)/Lyra (Zheng et al., 2024) |
| | PISA | 71.0 | pass@300 + timeout of 500s | Magnushammer (Mikuła et al., 2024)+Thor(Jiang et al., 2022) |
| Metamath | set.mm (valid/test) | 81.2/72.4 | pass@32 | HTPS (Lample et al., 2022) |
| Mizar | MPTP2078 | 75.5 | timeout of 100s | NIAGRA (Fokoue et al., 2023) |

Table 2: Summary of state-of-the-art methods and their performance on several key datasets.

---

[1] NL: natural language. DSL: domain-specific language.

[2] We use the number of tokens for pre-training datasets and the number of entries (e.g., definitions, premises, theorems) for other tasks. "*" indicates the size of the testing/validation set.

[3] We list the main task of each dataset. AF: autoformalization. PS$_1$: premise selection. PG: proofstep generation. PS$_2$: proof search. PT: pre-training. TP: theorem proving (combination of premise selection, proofstep generation, and proof search).

