# OpenReview forum: "A Survey on Deep Learning for Theorem Proving"
_colmweb.org/COLM/2024/Conference — COLM_

### Official Review · Reviewer_FB9a · 2024-05-10

**Rating:** 8
**Confidence:** 3
**Ethics Flag:** 1

**Summary:**

This paper offers a literature review on theorem proving, with a focus on recent deep learning approaches. The paper begins with introducing the informal and formal formulations of theorem proving, followed by presenting five common tasks in theorem proving and existing methods developed for each task, and then summarizes datasets, evaluation metrics and state-of-the-art model performances. The paper concludes with a discussion of outstanding challenges and promising future directions.

In my opinion this paper has a lot of good value. For a novice, it would be a great entry point for them to learn about the research landscape in ML-based theorem proving. For more senior researchers, it can also help organize their knowledge obtained from the large and scattering volume of work.

**Reasons To Accept:**

Overall, I think readers of this paper can gain a comprehensive bird-eye view of the most important recent advances in ML-based theorem proving. As someone who worked in theorem proving, I personally learned a lot from this paper.
1. The paper is well structured and clearly written.
1. This review has very good coverage of notable papers in the field of learning-based theorem proving. For those cited papers that I am familiar with, the summary is generally accurate.
1. It is very refreshing that the paper summarizes theorem proving tasks into five categories: autoformalization, premise selection, proofstep generation, proof search, and others. From the myriad of papers, this structure makes things very clear and easy to understand. Within each task, more fine-grained clustering is made (e.g., formulating the task as classification vs retrieval vs language modeling, early attempts using CNNs/RNNs vs more recent approaches leveraging pretrained LLMs and in-context learning).

**Reasons To Reject:**

I wouldn’t say these are reasons to reject, they are more like suggestions to make the paper more clear. Happy to discuss further if there’s any misunderstanding.
1. The presentation of datasets (Figure 3) can be made richer by including additional details, such as dataset size, data source, and annotation type (e.g. theorems, proofs, annotation of premises, fine-grained proof artifacts), so readers who are looking for data can more easily locate the best resource for them. This might be best presented as a table.
1. In section 3.4 “proof search”, I’m not sure if I understand why search methods are discussed separately for ATP and ITP. As mentioned in this section, in both domains, classical search methods like best-first search and MCTS are shown to be applicable. Can you maybe clarify how search is different in APT and in ITP?
1. Nit: In section 5 “theorem proving” paragraph, the Thor method seems to be implied as a “tactic-based model with best-first search” and contrasted with LLM-based methods and search-based methods. However, Thor also employs language models and proof search. It seems to me the takeaway here can be: using powerful LLMs can help, and using advanced search on top can also help. Some nuances could use some adjustment to make the story clearer here.

---

> ### Author Rebuttal · Authors · 2024-05-28
>
> Thank you for the positive review and valuable suggestions!
>
> > Presentation of datasets
>
> We appreciate your suggestion to enrich the presentation of datasets. We will incorporate a detailed table in the appendix that lists the existing theorem-proving datasets, including information such as language, size, source, and other relevant notes. This will enhance the usability of our survey for readers seeking specific data resources.
>
> > Proof search in Section 3.4
>
> While classical search methods like best-first search and MCTS are indeed applicable in both ATPs and ITPs, there are nuanced differences in how these methods are integrated with neural networks and their respective tasks. In ATP, search methods are paramount, often utilizing neural networks like RNNs and GNNs for embedding, and treating the process as a classification task for making branching decisions. In ITP, tactic prediction models are also crucial, allowing simple search methods like BFS/DFS to yield effective results. Moreover, existing proof search approaches for ITPs often model the task as language modeling, using language models to predict the proof length or success of a branch as the value function during the search. We will elaborate on these distinctions in the revised version.
>
> > Theorem proving in Section 5
>
> Thank you for pointing this out for better clarification. We will revise this paragraph accordingly to reflect the contributions of Thor and to clarify that both powerful LLMs and advanced search methods can significantly enhance theorem proving.
>
> We appreciate your insightful feedback and will address these points thoroughly in our revised paper. If you have any further suggestions or questions, please let us know!

---

> > ### Comment · Reviewer_FB9a · 2024-06-03
> >
> > Thanks for your response. I will keep the current scores. Great work!

---

### Official Review · Reviewer_pyKY · 2024-05-11

**Rating:** 8
**Confidence:** 4
**Ethics Flag:** 1

**Summary:**

This paper presents a comprehensive survey of deep-learning related exploration in the theorem proving. The survey features a summary of related tasks, datasets and evaluation metrics. At the end, the authors also discuss the challenges and the promising avenues for future exploration.

**Questions To Authors:**

I don't have any specific questions. The followings are merely some comments:
- page 2, 'saturation-based theorem provers'/'tableau-based', might be worth mentioning SAT/SMT-based approaches, which probably play the most important role in existing ATPs.
- page 4, 'premise selection': despite the research in deep-learning-based premise selection, most of the working systems are still based on heuristics. This is especially true for ATPs. In exceptions like LeanCopilot we usually don't do fast proof explorations. The trade-off between efficiency, engineering difficulty, and prediction accuracy could be a topic worth discussion.
- page 8, evaluation: Another (general) challenge in this field is the data contamination during the pretraining stage. This is also the reason that the Graph2Tac paper (https://arxiv.org/pdf/2401.02949) chose not to pretrain their LM baseline. More discussion is probably needed here.

**Reasons To Accept:**

The paper is well structured and comprehensive enough to include all the relevant papers I am aware of in the field.

**Reasons To Reject:**

I am not exactly sure about COLM's criteria for a survey paper. The main downside of this paper is the potential overlap of another survey paper published last year (https://arxiv.org/pdf/2212.10535), though I can see some of the latest work have been added.

---

> ### Author Rebuttal · Authors · 2024-05-28
>
> Thank you for the positive review and valuable feedback! We hope to address your comments in detail as follows:
>
> > Related survey paper
>
> Despite the mentioned [paper](https://arxiv.org/pdf/2212.10535) including some related works about theorem proving (~25 papers), the focus of that paper is primarily on general mathematical reasoning, particularly math word problems, rather than specifically on theorem proving (see their Table 8 for details). It includes very few papers on ATPs, autoformalization, or proof search, and it lacks our classification hierarchy as well as coverage of the most up-to-date methods involving advanced LLMs such as GPT-4. In contrast, our paper presents a comprehensive survey of deep learning approaches specifically for theorem proving, highlighting the latest advancements and unique contributions in this field.
>
> > Reference with SAT/SMT-based approaches in related work
>
> Thanks for the suggestion! We also acknowledge the importance of SAT, especially SMT solvers for ATP. Due to the page limit, we did not include them initially. We will add these approaches in our updated paper.
>
> > The trade-off between efficiency, engineering difficulty, and prediction accuracy
>
> As discussed in Section 6.1 (Evaluation and Human-AI interaction), we agree that there is a trade-off, especially between the efficiency and efficacy of theorem proving. This is also true for premise selection. We will expand this discussion to include more about premise selection in the revised version.
>
> > Discussion about data contamination
>
> Thank you for bringing this to our attention. We agree that data contamination (leakage) may occur, especially when using advanced LLMs like GPT-4, which may pre-train on existing testing data. We will add more discussion on this issue in our revision.
>
> We appreciate your insightful feedback and are committed to addressing these points comprehensively. If you have any further suggestions or questions, please let us know!

---

> > ### Comment · Reviewer_pyKY · 2024-06-05
> >
> > Thanks a lot for your response. I will increase my score.

---

### Official Review · Reviewer_UZhc · 2024-05-12

**Rating:** 8
**Confidence:** 4
**Ethics Flag:** 1

**Summary:**

This paper surveys the field of applying deep learning methods to theorem proving. An extensive summary of 170 papers is made on several tasks such as autoformaliztion, premise selection, proofstep generation, proof search and other tasks such as automated conjecturing. A taxonomy on the datasets involved in automated theorem proving is also made in a proper and comprehensive manner. Finally, a review of evaluations is given and a discussion on the future challenges is made, with an emphasis on data scarcity, evaluation and human-ai interaction.

**Questions To Authors:**

see above

**Reasons To Accept:**

* The summary is very well-structured and well-suited according to the types of tasks and datasets.
* The authors have good understanding of the general tendency of the field and its future. The challenges in Section 6 include important directions such as data scarcity.
* The writing is clear in general and easy to read

In general, automated conjecturing, automated theorem generation and reasoning data synthesis would be a very important direction in the future and it would be good to either have a separate subsection in Section 3 or emphasize it in 3.5 Other tasks.

**Reasons To Reject:**

* Several references are missing in some paragraphs:
   * “DT-Solver” in 3.3 after “Subsequent advancements”
   * “DT-Solver” in 3.4 after “Moreover, various works”
   * “MUSTARD” in 3.5 before “explore automated conjecturing”
* It would be nice to have some visualizations in the evaluation section for benchmarking and comparison purpose.

---

> ### Author Rebuttal · Authors · 2024-05-28
>
> Thank you for the positive review and valuable suggestions!
>
> Given the limited formal theorem and proof data, we agree that automated conjecturing (as well as theorem/proof synthesis) is a crucial direction for future research. It is highlighted in Sections 6.1 (Data Scarcity) and 6.2 (Conjecturing). We will emphasize these points further in Section 3.5 (Other Tasks). Besides, we will incorporate the suggested references and include two tables in the Appendix for better visualization: one listing the existing theorem-proving datasets and another benchmarking the performance of various theorem provers.
>
> Please feel free to follow up if you have further questions!

---

### Official Review · Reviewer_XYZ8 · 2024-05-21

**Rating:** 8
**Confidence:** 4
**Ethics Flag:** 1

**Summary:**

This paper provides a comprehensive survey on deep learning approaches for theorem proving.
The survey includes a thorough review of existing tasks and methods, meticulous summary of available datasets and strategies for data generation, a detailed analysis of evaluation metrics and the performance of state-of-the-art, and a critical discussion on the challenges and future directions.

**Questions To Authors:**

Please see the questions above.

**Reasons To Accept:**

The paper provides a very thorough summary and includes most of the related research works. I believe this paper provides a fundamental reference to the community. This paper is well-organized and very easy to follow. The discussions on current challenges and future directions are insightful.

**Reasons To Reject:**

Q1. References are missing in Section 2.1. For example, PrOntoQA [3] and EntailmentBank [4] are some demonstrations of informal theorem proving via natural language explanation.

Q2. Some search works may be missing. For example, ATG [1] introduces an automated theorem generation task and corresponding evaluation metrics to evaluate if a model can generate new theorems for a given theorem proving to achieve the shortest proof as human mathematicians. They construct the benchmark by synthesizing new theorems based on Metamath language and the “set.mm” library. This paper should be included in Section 4.2.

Q3. For proof generation via LLMs with several prompting methods, some of the methods also leverage the theorem prover feedback to prompt the LLMs to self-correct the proofs. For example, MUSTARD [2] uses the error messages from Lean prover to refine the proof generation.

Q4. Many references are not the newest version. For example, “Llemma: An Open Language Model for Mathematics”, “Holist: An environment for machine learning of higher-order theorem proving”, “Lego-prover: Neural theorem proving with growing libraries” are all published in conference proceedings. The authors need to carefully check the reference list to make sure the citations are up to date.


[1] Xiaohan Lin, Qingxing Cao, Yinya Huang, Zhicheng Yang, Zhengying Liu, Zhenguo Li, Xiaodan Liang (2024). ATG: Benchmarking Automated Theorem Generation for Generative Language Models. 2024 Annual Conference of the North American Chapter of the Association for Computational Linguistics (NAACL 2024 Findings).

[2] Yinya Huang, Xiaohan Lin, Zhengying Liu, Qingxing Cao, Huajian Xin, Haiming Wang, Zhenguo Li, Linqi Song, Xiaodan Liang (2024). MUSTARD: Mastering Uniform Synthesis of Theorem and Proof Data. The Twelfth International Conference on Learning Representations (ICLR 2024).

[3] Abulhair Saparov, He He (2023). Language Models are Greedy Reasoners: A Systematic Formal Analysis of Chain-of-Thought. The Eleventh International Conference on Learning Representations (ICLR 2023).

[4] Bhavana Dalvi, Peter Jansen, Oyvind Tafjord, Zhengnan Xie, Hannah Smith, Leighanna Pipatanangkura, Peter Clark (2021). Explaining Answers with Entailment Trees. Proceedings of the 2021 Conference on Empirical Methods in Natural Language Processing (EMNLP 2021).

---

> ### Author Rebuttal · Authors · 2024-05-28
>
> Thank you for your positive review and valuable feedback! In the following, please let us address the concerns and questions you've pointed out.
>
> > References about informal theorem proving via natural language explanation
>
> We appreciate your suggestion regarding the inclusion of PrOntoQA and EntailmentBank. We will add these references in our revised version to enrich the discussion on informal theorem proving via natural language explanations.
>
> > Missing references and reference versions
>
> Thank you for bringing these issues to our attention. We also noticed several recent works (e.g., ATG) published after our initial submission. We will incorporate these references and ensure our discussion reflects the latest advancements. Additionally, we will carefully check and update all references to ensure they are the most current versions.
>
> > Proof generation via LLMs and prompting methods
>
> The MUSTARD framework's focus on synthesizing theorem and proof data is primarily discussed in Section 4.2 (Data Generation). We will further elaborate on proof generation methods that leverage theorem prover feedback for self-correction, as seen in MUSTARD. Additionally, we will expand our discussion to include more advanced LLM prompting methods in proof generation in our revision.
>
> We hope these revisions will address your concerns and enhance the comprehensiveness of our submission. Thank you again for your constructive feedback!

---

> > ### Comment · Reviewer_XYZ8 · 2024-06-04
> > **Thanks for the response.**
> >
> > Thanks for the response! I will keep my rating.

---

### Decision · Program_Chairs · 2024-07-10

**Decision:**

Accept

**Comment:**

This survey paper on deep learning for theorem proving is impressively comprehensive and received uniformly positively assessments from reviewers. It's not easy to pack a comprehensive review into a 9-page conference paper the way the authors did. It's also notable that all reviewers brought up important, substantive points in their reviews, that the authors responded succinctly and clearly to all of them, and that all reviewers felt the author response was satisfactory. This is a model of a successful review–response process on both the reviewers' and the authors' parts. The paper should certainly be accepted.